# Genomic adaptations in information processing underpin trophic strategy in a whole-ecosystem nutrient enrichment experiment

Jordan G Okie[1]*, Amisha T Poret-Peterson[2], Zarraz MP Lee[3], Alexander Richter[4], Luis D Alcaraz[5], Luis E Eguiarte[6], Janet L Siefert[7], Valeria Souza[6], Chris L Dupont[4], James J Elser[3,8]

[1]School of Earth and Space Exploration, Arizona State University, Tempe, United States; [2]USDA-ARS Crops Pathology and Genetic Research Unit, Davis, United States; [3]School of Life Sciences, Arizona State University, Tempe, United States; [4]J Craig Venter Institute, La Jolla, United States; [5]Departamento de Biología Celular, Facultad de Ciencias, Universidad Nacional Autónoma de México, Mexico City, Mexico; [6]Departamento de Ecología Evolutiva, Instituto de Ecología, Universidad Nacional Autónoma de México, Mexico City, Mexico; [7]Department of Statistics, Rice University, Houston, United States; [8]Flathead Lake Biological Station, University of Montana, Polson, United States

*For correspondence:
jordan.okie@asu.edu

Competing interests: The authors declare that no competing interests exist.

**Abstract** Several universal genomic traits affect trade-offs in the capacity, cost, and efficiency of the biochemical information processing that underpins metabolism and reproduction. We analyzed the role of these traits in mediating the responses of a planktonic microbial community to nutrient enrichment in an oligotrophic, phosphorus-deficient pond in Cuatro Ciénegas, Mexico. This is one of the first whole-ecosystem experiments to involve replicated metagenomic assessment. Mean bacterial genome size, GC content, total number of tRNA genes, total number of rRNA genes, and codon usage bias in ribosomal protein sequences were all higher in the fertilized treatment, as predicted on the basis of the assumption that oligotrophy favors lower information-processing costs whereas copiotrophy favors higher processing rates. Contrasting changes in trait variances also suggested differences between traits in mediating assembly under copiotrophic versus oligotrophic conditions. Trade-offs in information-processing traits are apparently sufficiently pronounced to play a role in community assembly because the major components of metabolism—information, energy, and nutrient requirements—are fine-tuned to an organism's growth and trophic strategy.

## Introduction

Traits that influence the informational underpinnings of metabolism may be crucial to performance and community assembly but ecologists have largely focused on the proximal energetic and stoichiometric features of metabolism (*Leal et al., 2017*; *Sibly et al., 2012*; *Sterner and Elser, 2002*). Organisms must be able to store, copy, and translate the information contained in genetic material. And, they must be able to update their transcriptome and proteome adaptively in response to altered environmental conditions. For an organism to grow and reproduce rapidly, rates at every step of the metabolic network must be sufficiently high such that no single step is unduly rate-limiting, including the information processes that underpin biosynthesis and regulate metabolic networks. This necessary integration of functions is a hallmark of all organisms.

The structure and size of the genome affect the rate, efficiency, and robustness of the information processes that support metabolism, growth, and reproduction (Appendix 1). There are necessary tradeoffs in the costs and benefits of these features (Appendix 1; see also *Smith, 1976*), which should consequently make individual organisms more competitive in and better-suited to only particular ranges of growth and trophic conditions (*Roller and Schmidt, 2015*). Organisms that are best suited to compete in environments where resources are abundant (copiotrophs) must have the capacity for intracellular rates of information processing that are sufficiently high to support high rates of metabolism and reproduction. However, maintaining the genomic and structural capacity for rapid growth is costly, potentially placing copiotrophic taxa at a disadvantage in stable, nutrient-poor environments where growth rates are chronically slow (*Giovannoni et al., 2014*). Oligotrophic environments may thus instead favor organisms (oligotrophs), which have information processing machinery that is less costly to build, maintain, and operate, thereby increasing resource use efficiency and growth efficiency (*Koch, 2001*; *Roller and Schmidt, 2015*). Genomic traits that affect the rates and costs of biochemical information processing within cells can thus influence the degree to which an organism is optimized for oligotrophy versus copiotrophy.

This oligotrophic-copiotrophic strategy continuum is reminiscent of the classic slow-fast life history continuum (*Stearns, 1992*), of classical *r/K* selection theory (*MacArthur and Wilson, 2001*; *Pianka, 1972*; *Pianka, 1970*), and of their subsequent developments dealing with the evolution of interspecific variation in rates of resource use, mortality, growth, and reproduction (e.g., *Dobson, 2012*; *Grime and Pierce, 2012*; *Krause et al., 2014*; *Sibly and Brown, 2007*). In conjunction with research on the role of functional traits and niches in shaping the assembly of communities (e.g., *Fukami et al., 2005*; *Litchman and Klausmeier, 2008*; *McGill et al., 2006*; *Okie and Brown, 2009*; *Roller and Schmidt, 2015*), this work suggests that traits that are associated with the biological rates and efficiencies of resource use, growth, and reproduction play an important role in community assembly. It is unclear, however, whether the traits that are specifically related to rates and costs of biochemical information processing have sufficiently pronounced tradeoffs or physiological effects to play an important role in the evolutionary ecology of organisms and in the assembly of communities, although there are some promising indications (*Dethlefsen, 2004*; *Roller et al., 2016*).

Here, by coupling metagenomic analysis with a trait-based framework that synthesizes theory and hypotheses from genomics, ecology, and evolutionary cell biology, we investigate the role that several universal genomic traits play in determining the response of a planktonic microbial community to nutrient enrichment in a whole-ecosystem experiment. To date, relatively few studies have coupled metagenomics with a trait-based framework to clarify the drivers of community assembly (*Burke et al., 2011*; *Chen et al., 2008*; *Mackelprang et al., 2011*; *Raes et al., 2011*). Even fewer (indeed, none that we are aware of) have deployed such approaches in the context of whole-ecosystem experimentation to test ecologically relevant hypotheses under field conditions. We focus on a set of four information-processing traits that are hypothesized to affect the ability of organisms to obtain the high maximum growth rates necessary for thriving in copiotrophic environments, or their ability to reduce the energetic and resource requirements necessary to persist under nutrient-poor conditions:

1. *Multiplicity of genes essential to protein biosynthesis.* Copiotrophs are predicted to have higher copy numbers of rRNA operons and tRNA genes, because higher numbers of these genes increase their maximum overall transcription rates, helping maintain higher abundances of ribosomes (which are constructed from rRNA and proteins) and tRNAs. In turn, the larger pools of ribosomes and tRNAs facilitate the increased translation rates of protein synthesis necessary to achieve high growth rates (*Condon et al., 1995*; *Higgs and Ran, 2008*; *Rocha, 2004*; *Roller et al., 2016*). However, higher numbers of these genes incur costs, potentially putting organisms that have more copies of rRNA and tRNA genes at a disadvantage under low-resource conditions. In particular, more DNA has to be synthesized, maintained, and regulated, and there may be an increased risk of transcribing overly large phosphorus-rich pools of rRNA and tRNA, increasing phosphorus requirements (*Elser et al., 2003*; *Elser et al., 1996*; *Godwin et al., 2017*; *Makino et al., 2003*) and reducing growth efficiency (*Roller et al., 2016*).

2. *Genome size.* Organisms that have smaller genomes are predicted to do better in stable and oligotrophic environments, as they require fewer resources (such as phosphorus) to maintain

and replicate their genomes, have higher carbon-use efficiency (*Saifuddin et al., 2019*), and have smaller cells with increased surface-area-to-volume ratios that facilitate resource uptake (*Giovannoni et al., 2014*). By contrast, organisms that have larger genomes should do better in complex or copiotrophic environments, where they can take advantage of their typically higher intrinsic growth rates (*DeLong et al., 2006*) and their more diverse and flexible gene and metabolic networks (e.g., *Konstantinidis and Tiedje, 2004*; *Maslov et al., 2009*; *Saifuddin et al., 2019*) to facilitate substrate catabolism and to respond more rapidly to feasts following famines.

3. *GC content*. Genomic GC content (the percentage of DNA composed of the nucleotide bases guanine (G) and cytosine (C)) varies greatly among taxa. The reasons for this variation are controversial, as multiple different selective and neutral forces may be operating (*Bentley and Parkhill, 2004*; *Hildebrand et al., 2010*). However, researchers have proposed that G and C have higher energy costs of production and more limited intracellular availability compared to A and T/U (*Rocha and Danchin, 2002*). In addition, DNA and RNA that have a higher GC content have more nitrogen (*Bragg and Hyder, 2004*). Thus, lower genomic GC content may be favored in oligotrophic environments, whereas the metabolic and resource-sparing benefits of low GC content should be less consequential in resource-rich environments, leading to a relaxed role for GC content in the ecology and evolution of copiotrophs.

4. *Codon usage bias*. Eighteen of life's 20 proteinogenic amino acids can be encoded in the genome by more than one of life's 61 different proteinogenic codons (nucleotide triplets), leading to redundancy in the genetic code. However, these synonymous codons have different kinetic properties, including different translation rates and probabilities of mistranslation (*Higgs and Ran, 2008*). In highly expressed genes that are essential to growth (such as genes encoding ribosomal proteins), there should be increased selection for biasing the usage of certain synonymous codons over others in order to increase the accuracy and speed of translation, especially in organisms that have fast growth rates (*Hershberg and Petrov, 2008*; *Higgs and Ran, 2008*; *Plotkin and Kudla, 2011*; *Vieira-Silva and Rocha, 2010*). We thus predict that copiotrophic environments should favor organisms that have higher codon usage bias in their ribosomal protein genes, whereas codon usage bias should play a relaxed role in oligotrophic environments.

Further details and background on these traits are provided in Appendix 1.

Research on the genomic traits described above has revealed correlations of these traits with growth and trophic strategy in a variety of eukaryotic and prokaryotic species (*Vieira-Silva and Rocha, 2010*) and started to unravel the mechanisms by which these traits influence fitness (Appendix 1), suggesting that they may play an important role in ecology (*Freilich et al., 2009*; *Lauro et al., 2009*; *Weider et al., 2005*). However, the work published to date has tended to look at only one or two of these traits at a time (*Zeigler Allen et al., 2012*; *Raes et al., 2007*). More work is also required to help resolve incongruities in the literature, such as different views on the evolutionary ecology of bacteria genome size (e.g., *DeLong et al., 2010* versus *Vieira-Silva and Rocha, 2010*) (Appendix 1). It is also necessary to develop a less fragmented understanding of the evolutionary and physiological ecology of these genomic traits (*Bentley and Parkhill, 2004*; *Freilich et al., 2009*; *Vieira-Silva and Rocha, 2010*) and their role in community assembly across a wide range of organisms and environments.

Importantly, most ecological studies of these traits have been based on studies of microbial isolates, comparative analyses, or sampling across environmental gradients (*Zeigler Allen et al., 2012*; *Foerstner et al., 2005*; *Freilich et al., 2009*; *Raes et al., 2007*; *Roller et al., 2013*; *Vieira-Silva and Rocha, 2010*). Experimental work examining their role in mediating the structure of communities under natural conditions is extremely limited. Given (1) the complexity of the biotic and abiotic interactions that shape communities, (2) that most microbial taxa are uncultivable in isolation (*Ho et al., 2017*), and (3) that community traits can have widely different responses to temporal/experimental versus geographic variation in abiotic variables (e.g., *Sandel et al., 2010*), direct experimentation in the field with complex communities is required to better establish the validity of inferences about these purported molecular adaptations for ecological dynamics in nature.

Our study site is Lagunita, an oligotrophic, highly phosphorus-deficient pond in Cuatro Ciénegas, a biological reserve in Mexico (*Lee et al., 2015*; *Lee et al., 2017*). Because of its strong nutrient limitation, this ecosystem offers a useful setting for a fertilization experiment to evaluate the role of information-processing traits in community assembly and in the trophic strategies of organisms. Our study is noteworthy as one of the first whole-ecosystem experiments to involve *experiment-level*

*replicated* metagenomic assessments of community response. If it is true that individual genomic features do indeed affect the ability of organisms to survive and reproduce as a function of nutrient availability, then collective measures of these features at the metagenomic level (which aggregates the genomes of all of the individuals constituting a community) should likewise exhibit these characteristics and should reflect the organisms' responses to experimental fertilization (*Krause et al., 2014*; *Wallenstein and Hall, 2012*).

## Results

Biomass and chlorophyll *a* concentrations increased substantially in response to nutrient enrichment (biomass—198% mean increase, *p*=0.009; chlorophyll *a*—831% mean increase, p=0.001; *Appendix 1—figure 1*), as did the ratio of phosphorus to carbon (P:C) in seston biomass (19.5% mean increase, *p*=0.014, *Figure 1A*). We observed changes in several components of the predicted genomic signatures of growth and trophic strategy (*Figure 1*). As the percentages of bacteria, Archaea, Eukarya, and viruses making up the community were not discernibly different between unfertilized

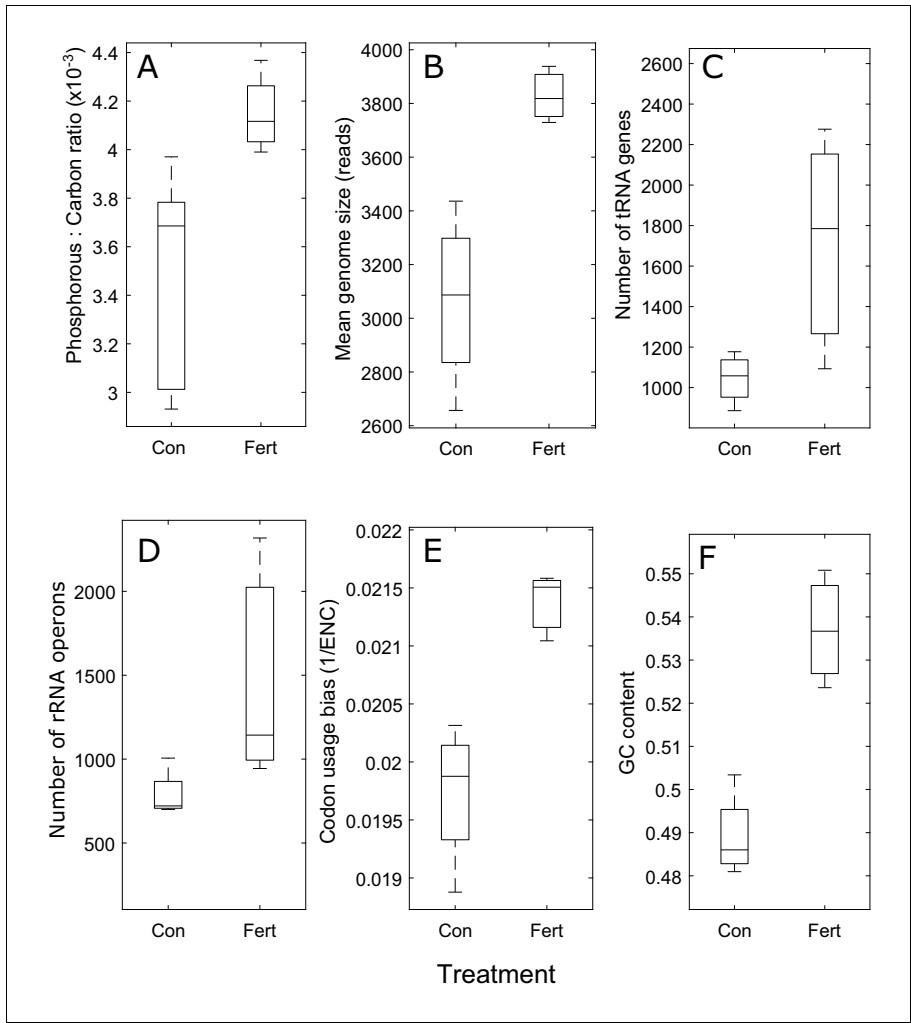

**Figure 1.** Community-level trait responses to nutrient enrichment. (A–F) As predicted, the mean value for each information-processing trait was higher in the fertilized treatment (Fert) than in the unfertilized treatment (Con). Because the faster-growing organisms generally have more P-rich ribosomes, seston P:C ratio also increased after fertilizer treatment (as shown in panel [A]). Boxes show 25/75% quantiles, center horizontal lines show medians, and vertical lines show the data range. In panel (E), a community's codon usage bias is the inverse mean effective number of codons (1/ENC) of a metagenome's ribosomal protein sequences, with higher 1/ENC values indicating increased codon usage bias.

and fertilized treatments ($p$=0.46, 0.37, 0.39, 0.21, respectively), these genomic changes reflected changes in the abundances of taxa at finer phylogenetic scales, in particular of bacterial taxa, which made up 94% of the metagenomes (see Appendix 1 and *Appendix 1—figures 1*, *2*, *3*, *4* for details). Furthermore, the genomic changes reflected widespread changes within the community—they were not driven by just a few specific populations, as 188 genera showed changes in abundance (with $p$<0.05) and no single genus dominated the community (the highest relative abundance of a taxon at the genus level was 4%). Consistent with predictions, the mean estimated genome size of bacteria was 25% higher in the fertilized treatment ($p$=0.011), with nutrient enrichment explaining 75% of the variation between samples in mean genome size (*Figure 1*). The GC content of open reading frames of DNA was 9.9% higher in the fertilized treatment: 54% compared to 49% in the unfertilized treatment ($p$=0.007, $R^2$ = 86%).

Genomic features that are indicative of adaptations for maintaining high rates of transcription and translation were also positively associated with fertilization. The per sequence occurrence rate of tRNA genes and the total number of tRNA genes per community were 93% and 64% higher, respectively, in the fertilized treatment than in the unfertilized treatment ($p$<0.001 and $p$=0.065 with $R^2$ = 53%, respectively). The residuals after regressing the log number of tRNA genes and the log total number of reads per sample to control for differences in sequencing depth were also higher in the fertilized treatment than in the unfertilized treatment ($p$=0.087, $R^2$ = 52%). Likewise, in the fertilized treatment, the per sequence occurrence rate of 16S rRNA genes and total number of rRNA operons per community were 119% and 86% higher, respectively ($p$<0.001 and $p$=0.096 with $R^2$ = 50%, respectively). The residuals after regressing log number of rRNA operons versus log total number of reads per sample were also higher ($p$=0.038, $R^2$ = 52%). Fertilization explained 65% of the co-variation in these two traits (number of rRNA operons and tRNA genes) along a single dimension, which was quantified by principal component analysis and provides a measure of protein synthesis capacity ($p$=0.031).

Consistent with predictions, nutrient enrichment also increased codon usage bias in ribosomal protein genes according to two measures of codon usage bias—the effective number of codons (ENC) and ENC' (*Figure 1E* and *Figure 2*). Values of ENC and ENC' for genes vary inversely with the level of codon usage bias: from 20, which signifies extreme codon usage bias in which one codon is used exclusively for each of the amino acids encoded by a gene, to 61, which represents the case in which the use of alternative synonymous codons is equally likely (no codon usage bias). The mean ENC and ENC' of ribosomal protein sequences detected in the metagenomes decreased with fertilization by 6.7% and 4.8%, respectively, indicating increased codon usage bias (ENC: $p$=0.018, $R^2$ = 66%; ENC': $p$=0.031, $R^2$ = 55%). Median ENC and ENC' were also lower in the fertilized

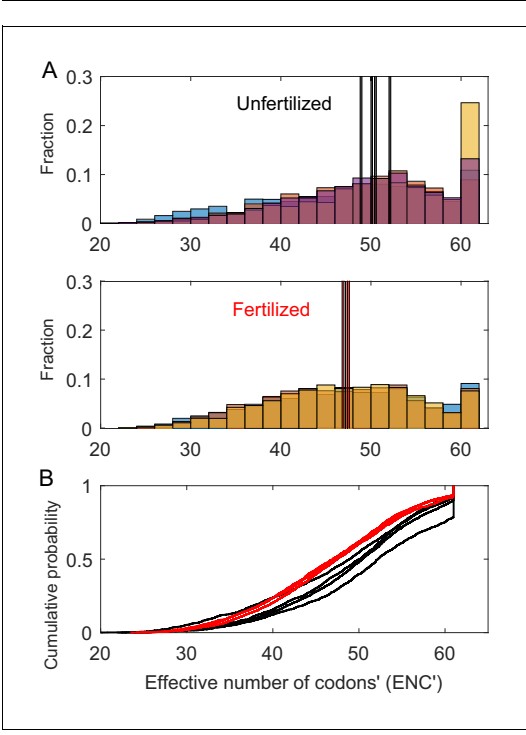

**Figure 2.** Histograms and cumulative distributions of the codon usage biases (ENC' values) of the ribosomal protein sequences in the metagenomes, with lower ENC' values indicating increased codon usage bias and thus increased speeds and/or accuracies of translation of ribosomes. (A) Histograms showing that the unfertilized communities have more sequences with nearly no detectable codon usage bias (values of 60–61), higher median ENC' values (indicated by the tall vertical lines), and typically fewer sequences with high codon usage bias (ENC' values from 20 to 40). Bar colors and their overlapping shades indicate different samples. (B) Cumulative distributions showing that, overall, the fertilized communities' distributions have fatter left-tails, so higher frequencies of ENC' values < 50 and thus more frequent codon usage bias. The Y-axis shows the proportion of ENC' values ≤ ENC' value designated by the plotted curves (fertilized = red curves, unfertilized = black curves).

treatment than in the unfertilized treatment ($p$=0.006, $R^2$ = 75% and $p$=0.010; $R^2$ = 75%), and Kolmogorov-Smirnoff tests indicated that the distributions of ENC and ENC′ in ribosomal protein sequences significantly differed between treatments (all $p$<0.01). In particular, as shown in *Figure 2A and B, a* greater frequency of sequences exhibited little-to-no codon usage bias (ENC and ENC′ values around 60–61) in the unfertilized communities (*Figure 2*), whereas in the fertilized treatment, there were notably higher numbers of ribosomal protein sequences with extremely high codon usage bias (ENC and ENC′ <~37).

As suggested by *Figures 1* and *2*, the variance in the P:C ratio and in the mean and median codon usage bias of communities (quantified by ENC′, the more powerful indicator of codon usage bias) substantially decreased with fertilization by around an order of magnitude or more—by factors of 8, 31, and 27, respectively (P:C ratio–$p$=0.058; mean ENC′–$p$=0.056; median ENC′–$p$=0.069; *Appendix 1—table 1*). Variance in genome size and mean and median ENC also decreased substantially by factors of 10, 10, and 6, respectively, but these responses are less certain (genome size–$p$=0.174; mean ENC–$p$=0.169; median ENC–$p$=0.275; *Appendix 1—table 1*). By contrast, variance in the log-transformed number of tRNA genes and rRNA operons substantially increased with fertilization, by 846% and 655%, respectively (rRNA–$p$<0.001; tRNA–$p$=0.013), whereas GC content variance did not appear to exhibit any change ($p$=0.56; *Appendix 1—table 1*).

Finally, we examined how well nutrient enrichment predicted the covariation of these genomic traits along a single principal component analysis (PCA) axis, which, according to our trait predictions, should quantify where a community's information-processing traits fall along an oligotrophy-copiotrophy strategy continuum. The single PCA axis captured 78% of the variance in the genomic traits for the communities, and nutrient treatment explained 86% of the variation in community genomic trait composition along this single axis of trophic strategy (general linear model [GLM]–$p$=0.004; *Figure 3*).

## Discussion

Our fertilization study demonstrated strong nutrient limitation in the Lagunita ecosystem, as manifested by increased biomass, chlorophyll content, and P:C ratios of plankton in the fertilized pond versus the unenriched mesocosms. Using metagenomics, we found strong differences in information-processing traits between the fertilized pond and unfertilized internal mesocosms, which agree with all five of our directional predictions. We are not aware of any obvious reasons or existing theoretical work to indicate that a difference in habitat size between treatments should lead to the observed trait differences. We thus interpret the observed genomic trait differences as resulting primarily from differences in the growth and nutrient conditions of the two treatments. However, a valuable future experiment could be to repeat the experiment using internal mesocosms for both the fertilized and unfertilized treatments in order to help to rule out potential mesocosm effects. Conservationists and ecologists should also have an involved discourse on whether it is worth conducting future research to verify our interpretation by performing much larger experiments

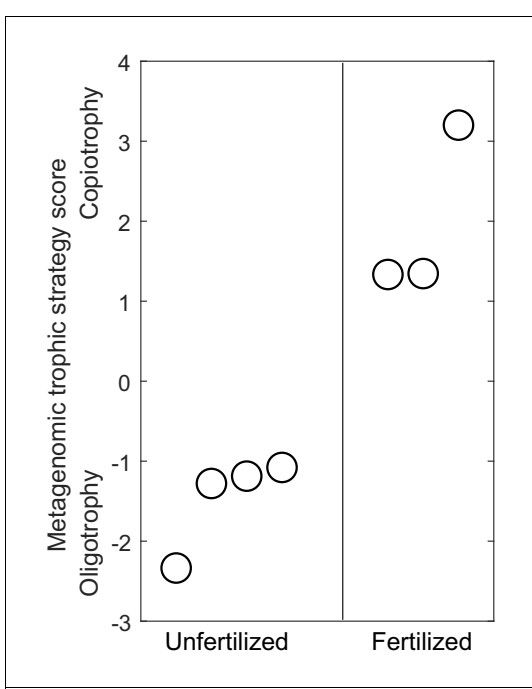

**Figure 3.** The principal component scores quantifying the position of metagenomes along a single dimension representing the oligotrophy-copiotrophy strategy continuum for information-processing traits. Higher scores on the vertical axis indicate communities whose information-processing traits are better suited for copiotrophy (increased maximum growth rate in resource abundant environments), whereas low scores indicate adaptation to oligotrophy (increased resource use efficiency in low nutrient environments). Nutrient enrichment explains 86% of the variance in metagenomes along this axis of growth and trophic strategy ($R^2$ = 86%, p=0.004). Within each treatment, circular symbols are spaced horizontally for visual clarity.

across multiple ponds (with wider environmental impacts).

We interpret these differences as primarily reflecting trait-mediated ecological dynamics—the differential success of lineages present within the pond and colonization by other lineages—rather than microevolutionary changes within populations. Our reasoning is that the relatively short time period of our experiment (32 days) encompasses a relatively low maximum number of generations of replication (Appendix 1), and so limited the opportunity for measurable genomic evolution to occur (e. g., *Lenski and Travisano, 1994*). Overall, our study suggests that ecologically significant phenotypic information about communities can be uncovered using a read-based approach (metagenomics) that leverages the relevant abundance and DNA characteristics of conserved genetic elements. Importantly, this approach avoids the biases associated with the massive gaps that are present in microbial taxonomic databases and cross-taxonomy assembly efficiencies (*Kunin et al., 2010*; *Sogin et al., 2006*; *Temperton and Giovannoni, 2012*; *Yooseph et al., 2010*).

## An oligotrophy-copiotrophy gradient in information-processing traits

Remarkably, nutrient enrichment explained 50% or more of the variation in each trait and 88% of the co-variation of these five traits along a single statistical dimension quantifying the oligotrophy-copiotrophy strategy continuum (*Figure 3*). The congruence of all trait responses with our predictions suggests that the effects of these genomics traits on the rates and costs of biochemical information-processing are sufficiently pronounced to play a role in community assembly.

We found that the fertilized community had genomic traits that were expected to augment translational capacity, whereas the unfertilized communities had genomic traits that lower the costs of biochemical information processing. For instance, the fertilized community had increased codon usage bias in ribosomal protein genes, which can improve the speed and accuracy of translation of ribosomal proteins by increasing the rate at which tRNAs bind to mRNA codons of ribosomal protein genes, as well as by possibly reducing the likelihood of mistranslation (*Higgs and Ran, 2008*; *Figures 1E* and *2*). The resulting increase in the rate of production of ribosomes supports the larger ribosomal pools undergirding higher overall rates of protein synthesis. The fertilized communities also had higher numbers of rRNA and tRNA genes. In conjunction with the observed increase in the biomass P:C ratio, this finding supports the 'Growth Rate Hypothesis'—that fast-growing cells should have higher P-contents that reflect higher concentrations of P-rich ribosomal RNA, which are maintained by increased copies of rRNA and tRNA genes (*Elser et al., 2000*).

Mean genome size was also higher in the fertilized pond, reflecting the need to encode a larger array of genes that are needed to support the expanded translational and catabolic capacity of faster growing cells, and possibly also reflecting the benefits of maintaining a streamlined genome that requires minimal resources for maintenance and replication in nutrient-poor conditions (e.g., *Giovannoni et al., 2005*; *Figure 1B*). Consistent with our prediction that low GC-content in oligotrophic taxa is a valuable resource conservation strategy, GC content was lower in the unfertilized communities (*Figure 1F*).

These results contrast with those of *Vieira-Silva and Rocha (2010)*, who found no significant interspecific correlation of GC content and genome size with generation time in a comparative study of bacteria (*Vieira-Silva et al., 2010*). The difference in results may reflect multiple factors, including: (1) the presence of confounding variables and pronounced statistical error (measurement and biological) inherent to interspecific microbial growth and trait databases that are collated from a variety of sources (e.g., ecosystems, conditions such as temperature), as in their study; (2) an underrepresentation in databases of the unculturable prokaryotic species that may comprise the majority of species in microbiomes, such as Lagunitas, and which typically have different evolutionary ecologies than culturable taxa (*Pande and Kost, 2017*; *Swan et al., 2013*). Such differences in results highlight the value of conducting in situ experiments.

Overall, however, our findings are largely consistent with observational studies of metagenomes and genomes along productivity gradients, and with comparisons across species varying in growth rate (*Zeigler Allen et al., 2012*; *DeLong et al., 2010*; *Foerstner et al., 2005*; *Lauro et al., 2009*; *Raes et al., 2007*; *Roller et al., 2013*; *Swan et al., 2013*). Our experiment on communities in situ suggests that much of the variation in these genomic traits along productivity gradients or across species may similarly be attributed to effects of information-processing costs and rates on an oligotrophy-copiotrophy strategy continuum and related adaptive strategies, such as *r/K* selection and Grime's 'C-S-R Triangle' (e.g., *Grime and Pierce, 2012*).

## Variance changes in the genomic traits

We also observed changes in the variances of several community-level traits (*Appendix 1—table 1*). The direction of the changes that occur in response to enrichment differed among the traits, so the variance responses do not simply reflect potential for increased dispersal limitation in unfertilized mesocosms as compared to enriched pond samples (see 'Materials and methods'). Instead, all-else-being equal, the responses suggest that the strength of a trait's role in filtering the community differs between unenriched and enriched treatments, which could reflect differences in the degree to which a trait affects the costs versus the rates of information processing. In general, traits that have much more of an impact on the costs of information processing should play more of a role in oligotrophic environments, whereas traits that have more of an impact on the rates of information processing (e.g., probably codon usage bias) should play more of a role in copiotrophic environments.

The substantial reduction in variance levels in codon usage bias in response to fertilization are in agreement with this viewpoint: the evolution of low codon usage bias is primarily thought to reflect relaxed selection rather than positive selection for neutral (random) codon usage, although many complex issues remain to be resolved (*Hershberg and Petrov, 2008*). By contrast, the highly increased variance in rRNA operon and tRNA gene numbers in response to fertilization may indicate that the augmented P and N requirements of taxa that have high rRNA and tRNA gene copy numbers, which tend to have larger pools of ribosomes and other translation machinery, are particularly detrimental in oligotrophic conditions (*Stevenson and Schmidt, 2004*). In nutrient-rich conditions, on the other hand, some organisms may still do well even with fairly low numbers of these genes, for example by having multiple genome copies (e.g., *Mendell et al., 2008*) or other mechanisms that increase the concentrations and kinetics of RNA polymerase, aminoacyl-tRNA synthetase, rRNA and tRNA (e.g., *Yadavalli and Ibba, 2012*).

We also observed a 10-fold decrease in variance of mean genome size in the unfertilized mesocosms, implying that genome size has a more variable ecological role in oligotrophic ecosystems, although this change was only suggestive (p=0.17). On the basis of genome streamlining theory alone, which suggests that small genomes should be favored in oligotrophic conditions because of the favoring of cellular architectures that minimize resource requirements (*Giovannoni et al., 2014*; *Morris et al., 2012*), we would expect reduced variance in genome size in oligotrophic environments, in contradiction with our results. Consideration of metabolic scaling may provide the explanation. There is presumably an interspecific increase in active mass-specific metabolic rate and $r_{max}$ with genome size in bacteria (*DeLong et al., 2010*), so under abruptly enriched conditions, community members that have larger genomes may displace the complex and diverse original community by growing faster. By contrast, it appears that the effects of genome size on metabolic costs may be insufficient to play as strong a role in the evolutionary ecology and assembly of these oligotrophic communities. This suggestion agrees with others who have argued that the elemental costs of DNA play a negligible role in the evolutionary ecology of genome size (*Lynch, 2006*; *Mira et al., 2001*; *Sterner and Elser, 2002*; *Vieira-Silva et al., 2010*)—the smaller genomes found in many parasites, symbionts, and oligotrophs may instead reflect the mutational loss of genes as the result of relaxed positive selection for keeping non-essential genes (*Mira et al., 2001*).

Unlike the other information-processing traits, we observed no change in variance of GC content between treatments. Although we expected environmental filtering for low GC content in oligotrophic conditions, it is less clear why there may be filtering for high GC content in response to nutrient enrichment, pointing to a need for further investigations into the implications of GC content for microbial physiological and community ecology.

## Concluding remarks

The genomic traits studied here affect the costs and rates of biochemical information processing within cells, and all of these traits responded as predicted to nutrient enrichment. Cells proliferating under nutrient-enriched conditions had increased capacities for transforming and storing information, whereas those persisting in oligotrophic conditions had genomic traits associated with reduced costs for the information processes that underpin metabolism and reproduction. Optimizing trade-offs in the efficiency and capacity of information processing may thus play a vital role in the

evolutionary specialization of a microbe's cellular biology to the particular trophic conditions of an ecosystem. Information processing traits should be included in the development of trait-based theories and frameworks for microbial community ecology, as apparently all three core components of metabolism—information, energy, and material requirements and transformations—must be closely fine-tuned to the growth and trophic strategy of a microorganism.

# Materials and methods

## Study site description

The whole-ecosystem fertilization experiment took place in Lagunita, a shallow (<0.33 m) pond averaging 35,000 L in volume and roughly ~12 m x 4 m. It is adjacent to a larger lagoon (Laguna Intermedia from the Churince system) in the Cuatro Ciénegas basin (CCB), an enclosed evaporitic valley in the Chihuahuan desert, Mexico. Despite its aridity, the CCB harbors a variety of groundwater-fed springs, streams, and pools. Past research has also shown that these aquatic environments harbor a high diversity of unique microbiota (*Souza et al., 2018*; *Souza et al., 2006*), which have evolved under strong stoichiometric imbalance (high nitrogen (N):phosphorus (P) ratios) and prevalent ecosystem P limitation (*Corman et al., 2016*; *Elser et al., 2005*). Lagunita water is high in conductivity, dominated by $Ca^{2+}$, $SO_4^{2-}$, and $CO_3^{2-}$, and has an average molar TN:TP ratio of 122 indicative of strong P limitation, as previously demonstrated in this system during a mesocosm experiment completed in 2011 (*Lee et al., 2015*; *Lee et al., 2017*). During the summer season, the pond shrinks substantially and the surface water temperature increases.

## Experimental design

On 25 May 2012, prior to initiation of fertilization, five replicate enclosures were established in different parts of the pond; these were unenriched treatments that served as reference systems for comparison with the pond after enrichment. As in *Lee et al. (2015)* and *Lee et al. (2017)*, each unenriched mesocosm consisted of a 40 cm diameter clear plastic tube enclosing around 41 L (based on an average depth of 0.33 m at the time at which the mesocosms were installed). This volume fluctuated slightly during the experiment and decreased very slightly towards the end of the experiment because of evaporation (which decreased the pond volume by 1.4%). The mesocosms were fully open to the atmosphere and sediments. Each mesocosm's water column was gently mixed periodically during our regular sampling (described below). Thus, with exposure to both the air and the bottom sediments, the unenriched mesocosms were essentially cylindrical 'cross-sections' of the ecosystem. The 41 L volume is a typical size for an aquatic mesocosm (e.g., see review of 350 mesocosms by *Petersen et al., 1999*) and appropriate for microbial studies, encompassing in the order of 30 billion prokaryotic cells (estimated from our cell counts). See Appendix 1 for more details.

The fertilization procedure was based on a previous mesocosm experiment in Lagunita (*Lee et al., 2015*; *Lee et al., 2017*). Prior to the initiation of the experiment, a morphometric map of the pond was created, allowing us to estimate the pond's water volume and to adjust that volume estimate as water depth changed through the season. Based on the pond's volume, we fertilized to increase the $PO_4^{3-}$ concentration in the water by 1 μM (as $KH_2PO_4$). We also added $NH_4NO_3$ in a 16:1 (molar) N:P ratio with the added P. The soluble reactive phosphorus (SRP) concentration of the pond was then measured every 3–4 days, after which we added sufficient $KH_2PO_4$ to bring the pond's in situ concentration back to 1 μM, along with the appropriate amount of $NH_4NO_3$ to achieve a 16:1 molar ratio. Fertilizer was added by mixing fertilizer solution with ~2 L pond water and broadcasting the mixture into all regions of the pond.

We thus performed a sustained whole-ecosystem fertilization treatment, with replicate internal unfertilized mesocosms serving as reference systems. Whole-ecosystem manipulation assures that any experimental responses are ecologically relevant, because the manipulated system encompasses the full scale and scope of ecosystem processes that might modulate that response (*Carpenter, 1998*). Such a whole-ecosystem approach can be especially powerful when coupled to appropriate reference systems. Although our internal unfertilized mesocosms were smaller than the surrounding fertilized pond, we consider them to be pertinent references for investigating the role of genomic traits in community assembly under differing nutrient conditions for several theoretical, empirical, practical, and ethical reasons. Although replicate whole ponds for comparison would be

preferred, the availability of multiple ponds at Cuatro Ciénegas for such experimentation is extremely limited given the basin's arid nature. Indeed, true replication of whole-ecosystem manipulations is very rarely achieved, at least in aquatic ecosystems. We thus followed the recommendations of *Carpenter (1989)* and *Carpenter (1998)*, relying on the application of a strong experimental treatment at the ecosystem scale and informed by previous experimentation (*Lee et al., 2017*) together with replication of internal reference mesocosms to assess the impacts of nutrient fertilization. In this way, we maximized the ecological realism of our perturbation by applying it at the ecosystem scale, while retaining the ability to compare manipulated dynamics against a benchmark. We preferred a whole-ecosystem fertilization of the pond over fertilizing internal mesocosms within the pond because the smaller enclosures might have provided an artificial view of how the microbial community responds to a nutrient perturbation at the ecosystem scale. In addition, mesocosms cut off many sources of colonizing species (such as shore sediments/soils in ponds) that can contribute to community reorganization following a perturbation (e.g., *Thibault and Brown, 2008*). Since this study aimed to understand the role of genomic traits in the assembly of communities (not just the disassembly caused by extinctions), it was thus important to avoid inhibiting community responses driven by the colonization of species.

Given the enormous heterogeneity between communities and water chemistry from different sites within the area, as well as the ability of microbes to disperse between ponds, using internal reference systems rather than other whole ponds is arguably more informative as it avoids introducing confounding factors related to variation between ponds (such as contrasting microbial communities) and instead introduces just one potentially confounding factor, the difference in size between the unfertilized mesocosms and the fertilized pond. We thus consider our approach to be the scientifically appropriate one for this conservation area. The design is a natural first step from experiments in small, homogenous bottles or bags. Scaling such experiments across multiple ponds/lakes may be a future step for experimental metagenomic research but not a responsible current step for research in the ecologically sensitive Cuatro Ciénegas basin area.

## Field monitoring, sampling, and routine water chemistry

Following initiation of fertilization, the pond and internal unfertilized mesocosms were sampled every four days to monitor basic biogeochemical and ecological responses (see Appendix 1 for water chemistry sampling details). At the end of the experiment (32 days), we sampled for metagenomics: five water samples from the pond itself (fertilized treatment) and one water sample from inside each of the five unfertilized internal mesocosms. It is worth noting that, given the substantial seasonal changes in temperatures and water chemistry, we think that comparing metagenomic data from the pond pre-fertilization to those from the fertilized pond 33 days later would be an inappropriate approach for gaining insight into the effects of fertilization. Thus, we focus on comparing post-fertilization metagenome data with temporally matched data from the unfertilized mesocosms.

The water inside the mesocosm was gently stirred with a dip net prior to sampling. Sampling involved submerging a 1 L polycarbonate beaker just under the surface of the water. Microbes in the water samples were filtered onto sterile GF/F filters (0.7 μm nominal pore size, Whatman, Piscataway, NJ, USA), frozen immediately in liquid nitrogen, and held at <80˚ C until laboratory DNA extraction, purification, and sequencing. Given the 0.70 μm pore size, extremely small prokaryotes were not part of our metagenomes and so our results do not apply to these picobacterioplankton. If anything, their inclusion would augment predicted community-level trait responses to fertilization, as picoplankton are slow-growers, tend to do poorly in nutrient-rich waters, have small genomes, and so are likely to decrease in abundance in the fertilized treatment. Routine water chemistry methods were used, as in *Lee et al. (2015)* and *Lee et al. (2017)*.

## DNA extraction, sequencing, annotation, and phylogenetics

DNA was extracted using the MO BIO PowerWater DNA Isolation kit with a slight modification (increasing volume of PW1 solution to 1.5 mL). DNA yield and quality were assessed by PicoGreen assay (Appendix 1) and prepared for sequencing on Illumina MiSeq with 12 samples per v2 2 × 250 bp sequencing run. Raw reads were trimmed of barcodes, quality filtered, and rarefied to 100,000 sequences per sample (Appendix 1). For the quality filtering, we used the standard Qscore of 25. Two samples from the fertilized treatment and one sample from the unfertilized treatment were left

out of subsequent analyses because they had sequencing depths less than 20% of the rest of the samples (whose sequencing depth averaged $2.5 \times 10^6$ reads) and low-quality scores. Sequences were phylogenetically annotated using the Automated Phylogenetic Inference System (APIS) with default parameters, which is designed to optimize annotation accuracy (*Zeigler Allen et al., 2012*). PCoA was used to visualize a Bray-Curtis distance matrix of the APIS annotations using the R (*R Development Core Team, 2011*) package vegan (*Oksanen et al., 2016*). We also used the statistical package *edgeR* (*Robinson et al., 2010*) in *R* to identify the taxonomic groups at the genus, class, phylum, and domain levels that exhibited discernable changes in relative abundance and to calculate the p-value of these changes. *edgeR* is specifically designed for dealing with sequencing count data in which there are minimal levels of replication (*Robinson et al., 2010*).

## Trait bioinformatics

We used Phylosift (*Darling et al., 2014*) to annotate non-sub-sampled libraries, allowing us to count the number of bacterial and archaeal 16S rRNA genes in each sample, which averaged 2607. Counting 16S rRNA genes provides a means of evaluating the copy number of rRNA operons, because 16S RNA genes in prokaryotes are typically transcribed as part of a rRNA operon. Furthermore, the typical situation, at least in the genomes of cultivated organisms, is that each bacterial rRNA operon has a single 16S rRNA gene (*Grigoriev et al., 2012*). The program tRNAscan-SE v1.4 (*Lowe and Eddy, 1997*), which is specifically designed for recognizing tRNA genes, was used with the provided general tRNA model in order to count the number of tRNA genes in non-sub-sampled libraries. Variation in the total number of tRNA genes indicates variation in gene copy numbers, as the number of tRNA genes per genome is driven by variation in tRNA gene copy number rather than by tRNA diversity (*Higgs and Ran, 2008*). In '*Statistics*', we describe our method for ensuring that results were not sensitive to variation in DNA sequencing depth. Although the tRNAscan-SE approach that we employed did not distinguish between Bacteria, Archaea and Eukaryotes, the observed rarity of Eukarya and insignificant changes in the groups' relative abundances between treatments (see '*Results*') indicate that the tRNAscan-SE results, and data from our other bioinformatic analyses, reflect variation in prokaryotes, namely Bacteria (Archaea are very rare in CCB samples; *Lee et al., 2017*), rather than in Eukaryotes.

Bacterial genome sizes were estimated according to methods in *Zeigler Allen et al. (2012)* (Appendix 1). Briefly, length normalized core marker gene counts that were identified as bacterial by APIS were used to determine the number of genome equivalents in a sample. The total number of predicted proteins annotated as bacterial by APIS was then divided by the number of genome equivalents.

To quantify the degree of synonymous codon usage bias for each observed bacterial and archaeal ribosomal protein gene sequence, we first used Phylosfit (*Darling et al., 2014*) to identify the sequences and then used the program *ENCprime* to calculate two commonly used metrics, the effective number of codons (ENC) (*Wright, 1990*) and a related measure, ENC' (*Novembre, 2002*), for each sequence. ENC and ENC are relatively statistically well-behaved and insensitive to short gene lengths compared to other measures of codon usage bias (*Novembre, 2002*; *Wright, 1990*). ENC' also accounts for departures in background nucleotide composition from a uniform distribution and is therefore considered to provide a more powerful and reliable measure of codon usage bias (*Novembre, 2002*). Background nucleotide composition for each sample was considered to be the average nucleotide frequencies of all of the samples' reads.

## Statistics

We used *t*-tests assuming unequal variances to evaluate mean differences between treatments. For community-level genomic traits, these tests were one-tailed with the alternative hypothesis based on the predicted differences described in the 'Introduction'. General Linear Models (GLM) with treatment (unfertilized vs fertilized) as a fixed factor were used to determine the amount of variation ($R^2$) in traits that was explained by treatment. Numbers of tRNA genes and rRNA operons were log-transformed before these analyses to achieve normality. In order to account for potential effects of sequencing depth on the number of rRNA operons and tRNA genes, we also regressed the numbers against a sample's log total number of reads and then performed GLM analyses on the residuals

(ascertaining whether or not, for a given sequencing depth, samples from the fertilized treatment have higher numbers of these genes, that is, higher residuals).

We also employed single-tailed Poisson rate tests to examine differences in the numbers of rRNA operons and tRNA genes. The Poisson rate test is more powerful than the *t*-test for examining differences in the per sequence rate of occurrences of tRNA genes and rRNA operons between treatments. In the Poisson rate test, the sample size for each treatment was the number of DNA samples and the treatment's observation length was the mean number of reads per metagenome. Two-sided Bonett's tests were used to test for treatment effects on variance between samples of community-level traits. Kolmogorov-Smirnoff tests were conducted to evaluate whether or not within-sample distributions of the ENC and ENC′ values of ribosomal protein gene sequences differ between treatments.

Finally, we examined how well nutrient enrichment predicted the covariation (correlation) of these genomic traits along a single axis quantified by principal component analysis (PCA). Should nutrient enrichment explain substantial variation in the communities along this single axis, then the PCA values would provide a measure of molecular adaptiveness to oligotrophic versus copiotrophic conditions. We used median ENC′ as the measure of codon usage bias in the PCA. We calculated the principal component score of each metagenomic sample along the first dimension and used GLM analysis to determine how well fertilization explained variation in the communities' scores along this dimension. Before performing PCA, in order to give equal weight to each trait, variables were first standardized (*z*-scored) by subtracting means and dividing by standard deviations. Overall, we aimed to avoid overreliance on significance levels and *p*-values in judging scientific results (*Carver, 1993*; *Halsey et al., 2015*; *McGill et al., 2006*; *Rothman, 2016*), so we report *p*-values and effect sizes and let readers judge the significance of the results for themselves.

We focused on the community response of genomic traits to varying nutrient conditions, rather than on a detailed natural history of the phylogenetic composition of the community, not only because we are interested in interrogating metagenomic changes within a trait-based framework, but also for pragmatic reasons. There are multiple bioinformatic challenges to resolving precisely the phylogenetic composition of entire microbial prokaryotic communities and ensuring that the phylogenetic biases of various molecular methods do not differ between environments or growth conditions (*Kunin et al., 2010*; *Sunagawa et al., 2013*). For instance, there are massive gaps in prokaryotic taxonomic databases (*Rinke et al., 2013*; *Temperton and Giovannoni, 2012*), and metagenomes generated from low and high growth communities in oceans have different levels of taxonomy blindness (*Kalenitchenko et al., 2018*; *Sogin et al., 2006*; *Yooseph et al., 2010*). Also, DNA sequence assembly introduces a bias, as clonal populations with even low coverage assemble very well whereas high abundance populations with strain diversity will not assemble well.

## Acknowledgements

This study was conducted with financial support from NSF (DEB-0950175) and NASA (NAI5-0018) grants awarded to JJE, WWF-FCS Alliance to VS and LEE, NASA (NNA15BB034A) to CD, NSF (1536546) and NASA NAI (NNH05ZDA001C, NNH12ZDA002C, NNA08CN87A, NNA13AA93A) to JLS, and NASA (NNX16AJ61G) to JGO. This study was made possible with the sampling permit from Vida Silvestre-SEMARNAT, granted to V Souza (09762). We thank J Learned, J Corman, J Ramos, and E Moody for field assistance, the Cuatro Ciénegas community for its hospitality, S Vieira-Silva for advice and sharing of the codon usage bias analysis program *growthpred-v1.07*, and David Donoso, Detlef Weigel, and the reviewers for thoughtful feedback. This paper was written during a sabbatical leave that allowed LEE and VSS to spend time at the University of Minnesota in Peter Tiffin's laboratory and in Michael Travisano's laboratory, respectively, with support from scholarships from PASPA, DGAPA, UNAM.

## Additional information

### Funding

| Funder | Grant reference number | Author |
| --- | --- | --- |
| National Science Foundation | DEB-0950175 | James J Elser |

| National Aeronautics and Space Administration | NAI5-0018 | James J Elser |
|---|---|---|
| WWF-FCS Alliance | | Luis E Eguiarte Valeria Souza |
| National Aeronautics and Space Administration | NNA15BB034A | Chris L Dupont |
| National Science Foundation | 1536546 | Janet L Siefert |
| National Aeronautics and Space Administration | NNH05ZDA001C | Janet L Siefert |
| National Aeronautics and Space Administration | NNH12ZDA002C | Janet L Siefert |
| National Aeronautics and Space Administration | NNA08CN87A | Janet L Siefert |
| National Aeronautics and Space Administration | NNA13AA93A | Janet L Siefert |
| National Aeronautics and Space Administration | NNX16AJ61G | Jordan G Okie |

The funders had no role in study design, data collection and interpretation, or the decision to submit the work for publication.

## Author contributions

Jordan G Okie, Conceptualization, Software, Formal analysis, Investigation, Visualization, Methodology, Project administration; Amisha T Poret-Peterson, Software, Formal analysis, Investigation, Visualization; Zarraz MP Lee, Investigation; Alexander Richter, Data curation, Investigation; Luis D Alcaraz, Conceptualization; Luis E Eguiarte, Janet L Siefert, Valeria Souza, Conceptualization, Funding acquisition, Investigation; Chris L Dupont, Conceptualization, Data curation, Formal analysis, Supervision, Investigation; James J Elser, Conceptualization, Supervision, Funding acquisition, Investigation, Methodology, Project administration

## Author ORCIDs

Jordan G Okie https://orcid.org/0000-0002-7884-7688
Luis D Alcaraz http://orcid.org/0000-0003-3284-0605
Luis E Eguiarte http://orcid.org/0000-0002-5906-9737
Valeria Souza http://orcid.org/0000-0002-2992-4229

## Decision letter and Author response

Decision letter https://doi.org/10.7554/eLife.49816.sa1
Author response https://doi.org/10.7554/eLife.49816.sa2

# Additional files

## Supplementary files

• Source data 1. Data on the metagenomic traits and concentrations of seston chlorophyll a, phosphorus, nitrogen, and carbon in water samples from Lagunitas pond, Cuatro Ciénegas, Mexico.

## Data availability

Raw sequence data and metadata have been submitted to the NCBI Sequence Read Archive, accessible through BioProject PRJEB22811.

The following dataset was generated:

| Author(s) | Year | Dataset title | Dataset URL | Database and Identifier |
|---|---|---|---|---|
| J Craig Venter Institute | 2017 | Cuatro Cienegas Lagunita Fertilization Experiement | https://www.ncbi.nlm.nih.gov/bioproject/?term=PRJEB22811 | NCBI BioProject, PRJEB22811 |

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

**Appendix 1**

<div style="border-left: 4px solid #2d7ba8; padding-left: 1em;">

## Further background on the genomic traits

We have highlighted reasons for the choice of genomic traits used in this study in the introduction of the main text. Below, we provide more details and references on the relevance of these traits for the rates and costs of information processing and their role in community assembly and the trophic strategy of organisms.

### Copy number of highly expressed genes essential to biosynthesis

In order to grow quickly, organisms must be able to transcribe genes, translate mRNA, and synthesize proteins at a sufficiently fast rate. As ribosomes catalyze protein synthesis, increased capacity to replenish diminished ribosomal pools or to maintain large standing stocks facilitates growth. Thus, faster-growing organisms should benefit from increasing their number of rRNA operon copies (and so the 16S rRNA gene) in order to increase the rate of transcription of rRNAs, allowing the size and rate of turnover of the ribosomal pool to be larger and thereby preventing translation by the ribosomes from becoming the rate-limiting step in cellular growth.

Indeed, in Bacteria and Eukaryotes there are intraspecific and interspecific correlations between rRNA operon copy number per genome, which can vary by over an order of magnitude in bacteria (*Acinas et al., 2004*; *Větrovský and Baldrian, 2013*), and growth rate or generation time (*Condon et al., 1995*; *Gyorfy et al., 2015*; *Klappenbach et al., 2000*; *Lauro et al., 2009*; *Shrestha et al., 2007*; *Stevenson and Schmidt, 2004*; *Yano et al., 2013*). Thus microbes that are competitive in fast-growing environments are likely to have to have a greater number of copies of the rRNA gene operon.

Furthermore, because faster-growing organisms must have larger cellular quotas of rRNA, they should have higher phosphorus contents (*Elser et al., 2003*; *Elser et al., 1996*; *Makino et al., 2003*) as rRNA is phosphorus rich, and thus require more phosphorus-rich resources in order to build these rRNA pools. These observations, known as the 'growth rate hypothesis' (*Elser et al., 2000*), have received considerable empirical support from comparative studies, but research exploring their usefulness as a predictive trait for community ecology has been limited.

Likewise, since the concentrations of tRNAs affect the rate of protein translation, having more copies of tRNA genes, as well as a greater diversity of anticodons among the tRNA genes, may help a cell to maintain larger pools of tRNAs (as has been observed [*Higgs and Ran, 2008*; *Kanaya et al., 1999*]) and thereby maintain the faster translation rates required for faster growth rates. Indeed, the total number of tRNA genes (which, for example, varies from under 30 to over 120 per genome in bacteria) has been shown to be correlated negatively with generation time (*Higgs and Ran, 2008*; *Rocha, 2004*).

### Genome size

Genome size is a complex trait that affects multiple aspects of an organism's ecology, physiology, and molecular biology, making it both an important trait to study as well as a challenging one because its influence on ecology may differ between environments, organisms, and historical circumstances. Many microbiologists previously thought that there was no relationship between growth rate and genome size (*Mira et al., 2001*). Recent work suggests that among species growth rate and genome size may be positively correlated (*DeLong et al., 2010*), although this remains controversial and conflicts with some studies that may not have addressed confounding variables such as the effects of temperature on growth rate (*Vieira-Silva et al., 2010*; *Vieira-Silva and Rocha, 2010*). In addition, bacteria in oligotrophic zones of the oceans tend to have smaller genomes than bacteria in copiotrophic zones (*Zeigler Allen et al., 2012*; *Lauro et al., 2009*). Thus, organisms with larger genomes

</div>

may, all else being equal, do better under high growth conditions and so respond favorably to nutrient fertilization.

There are several non-mutually exclusive reasons why genome size may be correlated with growth rate and response to fertilization. For one, because DNA is P-rich, large-genome organisms require more phosphorus to maintain and replicate their genomes and so they have greater difficulty than small-genome species in obtaining sufficient amounts of P in oligotrophic, P-limited environments. Larger-genome species also tend to have larger cells, leading to, on average, decreased surface-to-area-volume ratios and a consequent disadvantage in obtaining sufficient nutrients (*Okie, 2013*). Genome size also affects the size, structure and function of the metabolic networks of organisms by allowing for a greater diversity of enzymes, a higher diversity of metabolic pathways, and enhanced metabolic multi-functionality, which may affect an organism's ability to take up a variety of substrates, the speed of resource uptake and transformation, and the yield of resource transformation (*DeLong et al., 2010*; *Maslov et al., 2009*). Finally, larger genomes allow organisms to encompass greater copy numbers of highly expressed genes and genes involved in protein translation machinery, such as tRNA and rRNA genes, which in turn allow for greater translation rates, as discussed above.

## Nucleotide base composition of DNA

The percentage of a genome's DNA composed of the nucleotide bases guanine and cytosine is known as a genome's GC content. Genome GC content varies widely across life, and there appears to be pervasive selection on the GC content of bacterial genomes, leading to selection for high GC content in some bacterial genomes (*Hildebrand et al., 2010*). The reasons for positive selection on GC content are contested, and it is likely that there are multiple different selective forces on GC content related to the niches and environmental conditions of different species. Because GC bonds have eight nitrogen atoms whereas AT bonds contain only seven, higher GC content genomes tend to have higher nitrogen content (*Bragg and Hyder, 2004*), so that higher GC content organisms have increased nutrient requirements for replication, DNA repair, and their mRNAs.

*Rocha and Danchin (2002)* found that parasitic and symbiotic bacteria have lower GC content than free-living bacteria, and proposed an explanation based on the biochemical details of nucleotide metabolism and differences in the energetic expense of GTP and CTP nucleotides versus ATP and UTP nucleotides. The explanation boils down to the difference resulting from increased selection by competition for scarce resources in parasites and symbionts. Here, we extend this explanation to include extremely oligotrophic genomes as similarly susceptible to these evolutionary forces as parasites and symbionts, and thus propose that an emergent consequence of these effects should be a positive association between GC content and growth rate. Suggestive support for this possibility is provided by the observation that GC content tends to be higher in the DNA of environmental samples from complex environments, such as soils, and from environments with high amount of nutrients, such as whale carcasses and copiotrophic waters, compared to the DNA of samples from simple and lower productivity environments, such as oligotrophic pelagic communities in the Sargasso Sea (*Zeigler Allen et al., 2012*; *Foerstner et al., 2005*; *Raes et al., 2007*).

## Codon usage bias

An amino acid can be encoded by multiple different codons (nucleotide triplets), but these synonymous codons have different kinetic properties, including different probabilities of mistranslation. In highly expressed genes that are essential for growth, such as ribosomal protein genes, there should be increased selection for biasing the usage of certain synonymous codons over others in order to optimize the accuracy and speed of translation, especially in organisms with fast growth rates.

As 18 of the 20 standard amino acids are encoded by two or more synonymous codons, genomes and genes may favor certain codons over others without altering translation

products. Different synonymous codons have different probabilities of mistranslation. Thus, the use of more accurately translated codons can be beneficial because mistranslation wastes energy, reduces the translation rate (all else being equal), and/or can cause cytotoxic protein misfolding. Codon usage bias (CUB) has been documented in Bacteria, Archaea, and Eukaryotes (*Satapathy et al., 2014*; *Subramanian, 2008*; *Vieira-Silva and Rocha, 2010*). The highly expressed genes of a genome, such as ribosomal protein genes, tend to have greater CUB (*Subramanian, 2008*; *Vieira-Silva and Rocha, 2010*). As CUB can alter the accuracy and speed of translation, this greater CUB presumably reflects selection for a greater translational accuracy and/or or efficiency (*Hershberg and Petrov, 2008*), which should increase rates of translation.

It follows from this positive effect of CUB on translation that faster-growing organisms should tend to have greater CUB, especially in genes that are highly expressed and essential for growth (such as ribosomal protein genes). Indeed, a negative relationship between an organism's generation time and the CUB of its ribosomal protein genes has been reported (*Subramanian, 2008*; *Vieira-Silva and Rocha, 2010*). A few studies have compared the CUB of metagenomes from different environments (*Roller et al., 2013*; *Vieira-Silva and Rocha, 2010*), but more work is required to clarify the role of CUB in community assembly.

## Additional methods

### Experiment
A small number of fish and larger aquatic macroinvertebrates (~1 cm or greater) were removed from the unfertilized mesocosms with a dip net before beginning the experiment. Such removals were necessary to ensure that enclosures did not experience unduly large stochastic disruption from animal activities resulting from differing numbers of large animals being trapped inside at unnaturally high densities. This would have confounded the difference in nutrient conditions between the enriched and unenriched treatments, undermining the purpose of the experiment. To the extent that such consumers augmented nutrient availability outside of the unenriched enclosures, then their removal from the enclosures would have amplified the nutrient enrichment contrast between the fertilized pond and the unenriched enclosures.

### Field monitoring and sampling
Water was sampled adjacent to and within each of the five internal mesocosms. Samples were filtered onto Whatman GF/C filters for analysis of chlorophyll (chl *a*) concentrations and passed through 0.2 μm polyethersulfone membrane filters (Pall Life Sciences, Port Washington, NY). Filtrate was used for analyses of nitrate ($NO_3$), ammonia/ammonium ($NH_{3/4}$), soluble reactive phosphorus (SRP), and total dissolved phosphorus (TDP). After 16 and 32 days of fertilization, water samples were also filtered onto pre-combusted Whatman GF/F filters for analysis of concentrations of C, N, and P in suspended particulate matter (seston particles). Unfiltered water samples were frozen for later analysis of total N (TN) and total P (TP) concentrations.

### Routine water chemistry
Chl *a* on GF/C filters was quantified fluorometrically using a TD-700 fluorometer (Turner Designs, Sunnyvale, CA) after 16–24 hr of extraction in cold absolute methanol (*Arar and Collins, 1997*). TN concentrations were measured using a Shimadzu TOC/TN analyzer. TDP and TP concentrations were measured using the same colorimetric method after persulfate digestion of the filtered or unfiltered samples, respectively. GF/F filters with seston were thawed, dried at 60°C and then packed into tin discs (Elemental Microanalysis, UK) for C and N analyses with a Perkin Elmer PE 2400 CHN Analyzer at the Arizona State University Goldwater Environmental Laboratory. Another set of dried GF/F filters prepared from the same water

samples was used for estimation of seston P content. These filters were digested in persulfate followed by colorimetric analysis for P.

## Calculations on the potential for evolutionary change during the experiment

For microorganisms with 2 hr minimum generation times—the approximate average minimum generation time of prokaryotic isolates in many data sets (e.g., *DeLong et al., 2010*; *Vieira-Silva and Rocha, 2010*)—a maximum of 384 generations of replication can occur in the 32 day period of the experiment, which is barely sufficient for much evolutionary change (*Lenski and Travisano, 1994*), especially for change in some of the genome-scale traits such as GC content and codon usage bias. Many of the microorganisms in natural ecosystems (that is, the unculturable taxa underrepresented in generation time datasets that make up the majority of prokaryotic communities) probably have minimum generation times greater than 2 hr, and under field conditions, they do not achieve or sustain their minimum generation times for the whole course of the experiment. Thus, we expect that in our experiment, populations experienced much fewer than 384 generations, providing very limited opportunity for evolutionary change to affect the genomic traits.

## Results – changes in taxonomic composition

The percentages of Bacteria, Archaea, Eukarya, and viruses making up the communities were not discernibly different between unfertilized and fertilized treatments (p=0.46, 0.37, 0.39, 0.21, respectively). For Bacteria and Eukarya, the percentages varied relatively little between samples within and across treatments ($R^2$ = 9.6% and 13.5%, respectively). For Archaea and viruses, the mean percentages were substantially different between treatments (mean percentages in the fertilized treatment were 43% and 46% lower, respectively, than those in the unfertilized treatment), but owing to the high within-treatment variance, treatment explained limited variation in the relative abundances of these domains ($R^2$ = 14% and 27%, respectively). The mean relative abundances in the unfertilized versus fertilized treatments were, respectively, 94% versus 93% for Bacteria, 6.0% versus 6.6% for Eukarya, 0.40% versus 0.23% for Archaea, and 0.05% versus 0.03% for viruses. Thus, Bacteria dominated the communities in both treatments. However, as expected, nutrient enrichment altered microbial community composition at a finer phylogenetic resolution, as indicated by the Principal Coordinates Analysis (PCoA) plot of community phylogenetic composition (based on the APIS analyses; *Appendix 1—figure 2*) and supported by statistical analysis: the PCoA scores from a two-dimensional analysis differed between treatments (first dimension—$R^2$ = 45.6%, *p*=0.096; second dimension—$R^2$ = 51.3%, *p*=0.070) and several taxonomic groups changed in abundance with fertilization (*Appendix 1—figures 3* and *4*, all *p*<0.01). Our results thus confirm the rarity of microbial eukaryotes in the pond and show modest changes in the group's relative abundance between treatments, indicating that the tRNA counts and our other bioinformatic analyses mostly reflect the response in Bacteria.

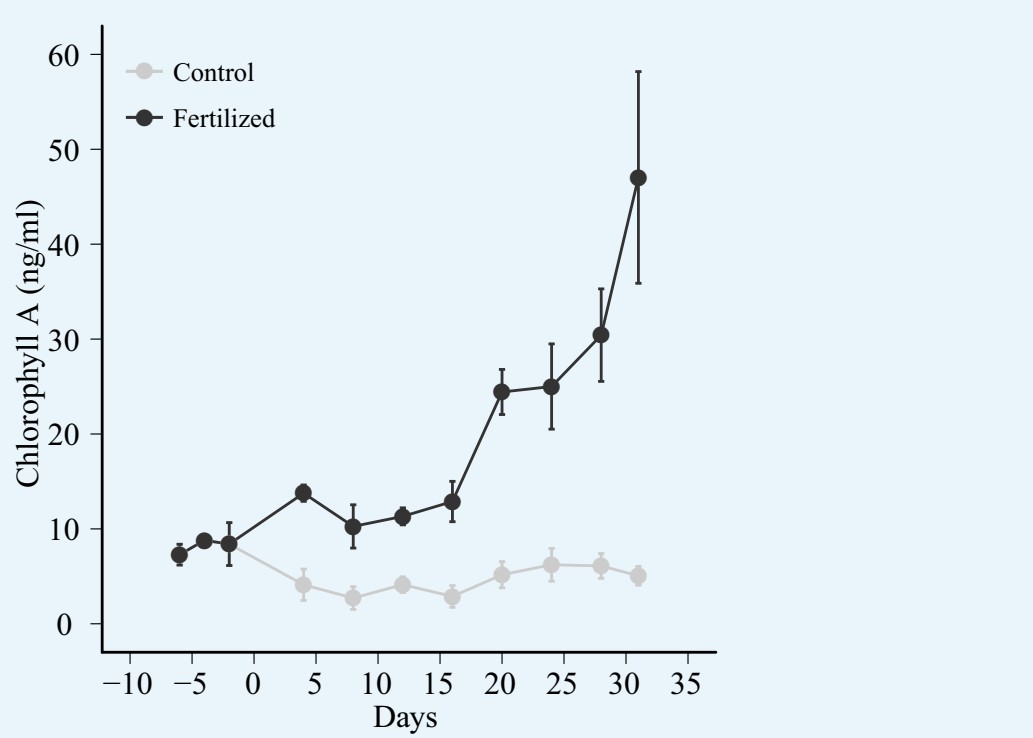

**Appendix 1—figure 1.** Chlorophyll a concentration increased with nutrient enrichment, whereas it remained relatively invariant in the unfertilized treatment.

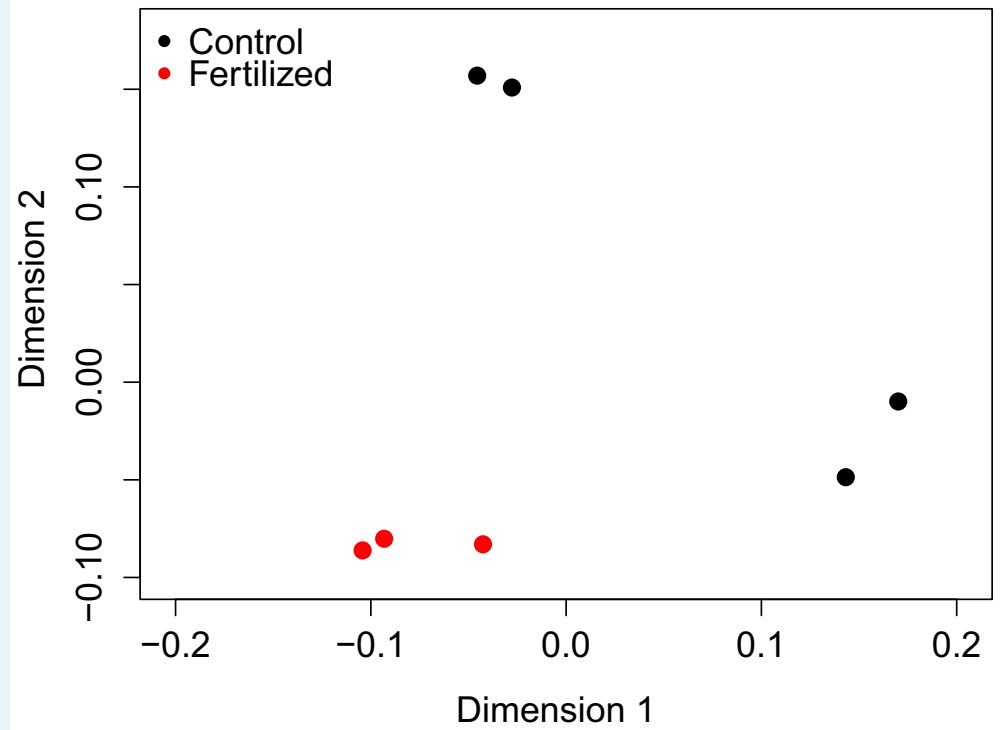

**Appendix 1—figure 2.** Principal coordinates analysis (PCoA) of the community phylogenetic structure inferred from the metagenomes shows that the phylogenetic composition of samples, as indicated by each point, is substantially different between unfertilized and fertilized treatments. Microbial community phylogenetic composition also varied notably within the unfertilized mesocosms, falling into two clusters driven by variation in the relative abundance

of Alphaproteobacterum, whereas enriched communities all shared relatively similar phylogenetic composition, indicating a convergence of effects of fertilization on community composition.

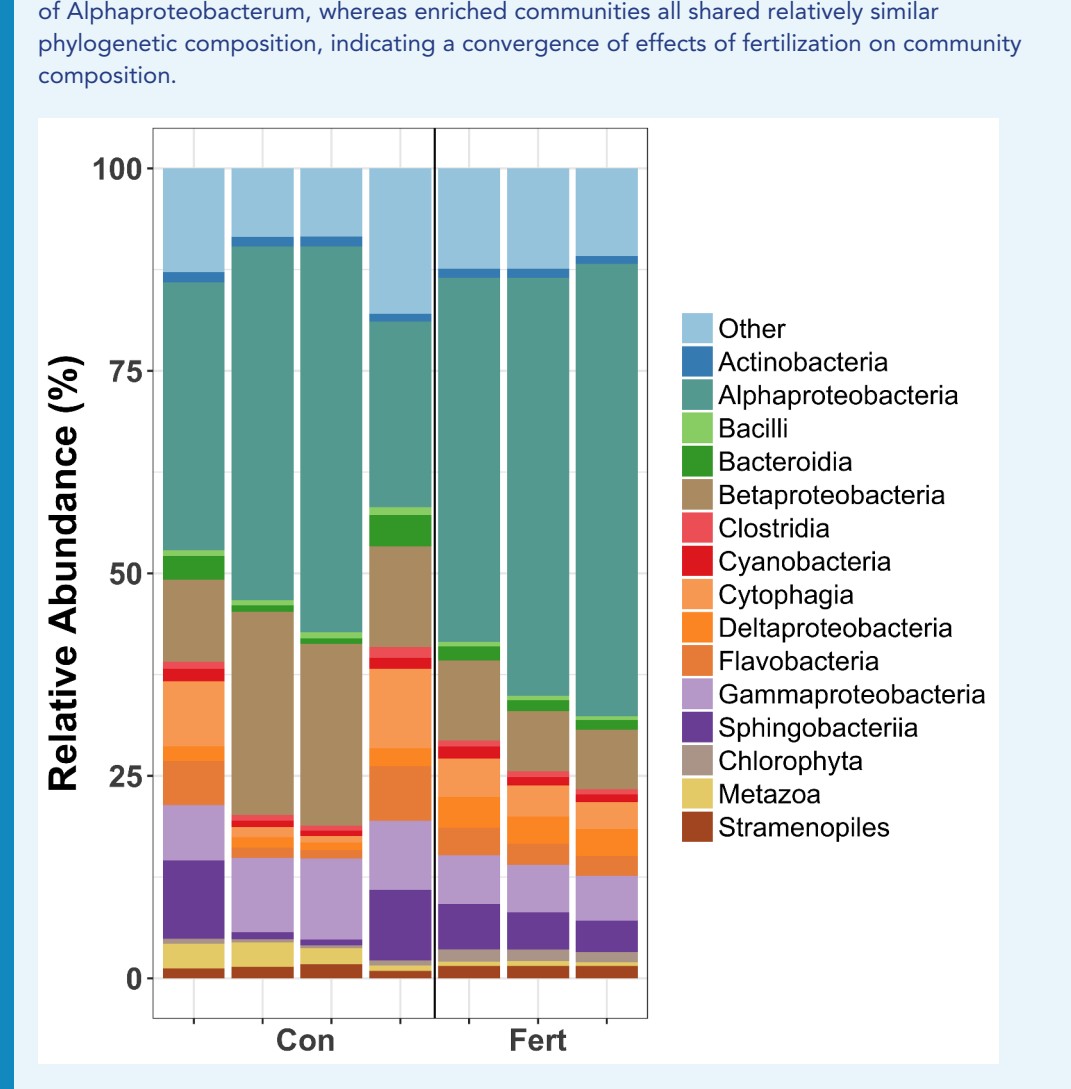

**Appendix 1—figure 3.** Relative abundances of taxonomic phyla in the samples from the unfertilized (Cont) and fertilized (Fert) treatments.

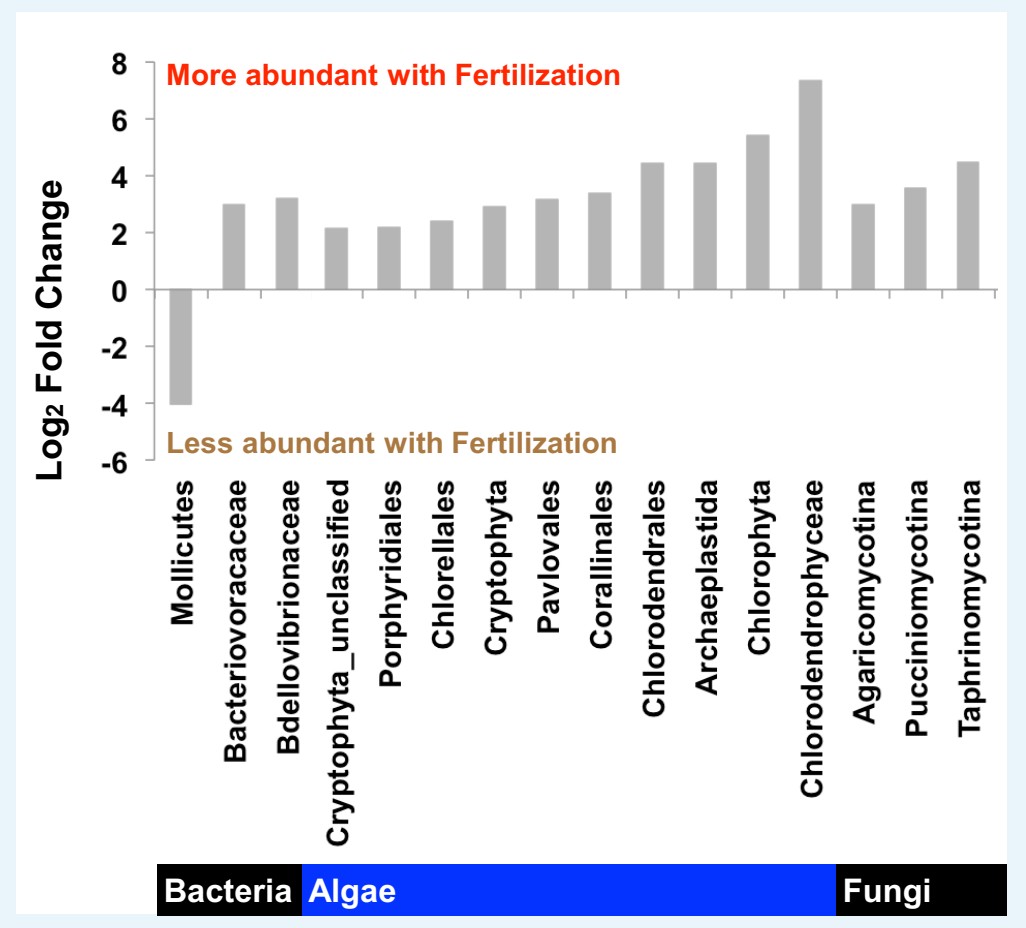

**Appendix 1—figure 4.** The significant relative abundance changes in taxonomic classes suggested by the statistical analysis of the phylogenetic composition determined by APIS. The statistical analysis is based on the statistical methods developed in the EdgeR package (*Robinson et al., 2010*). Note that less than 50% of reads could be annotated. All $p<0.01$.

**Appendix 1—table 1.** Differences in the variance of community-level traits between treatments.

| Community-level trait | Variance | | Ratio of variances | Bonett's test *p*-value |
|---|---|---|---|---|
| | Unfertilized | Fertilized | | |
| P:C ratio | $2.09 \times 10^{-7}$ | $2.06 \times 10^{-8}$ | 0.13 | 0.058 |
| Mean genome size | 105,315.4 | 10,984.8 | 0.1 | 0.174 |
| Log number of rRNA genes | 0.0056 | 0.0422 | 7.55 | <0.001 |
| Log number of tRNA genes | 0.0148 | 0.1396 | 9.46 | 0.013 |
| GC content | 0.0001 | 0.0002 | 1.9 | 0.56 |
| Mean ENC | 3.0554 | 0.3034 | 0.1 | 0.169 |
| Median ENC | 2.5861 | 0.413 | 0.16 | 0.275 |
| Mean ENC' | 2.5494 | 0.0824 | 0.03 | 0.056 |
| Median ENC' | 1.832 | 0.0671 | 0.04 | 0.069 |

