## [Decision Letter]

**Acceptance summary:**

All living things store the rules that shape their lives in a genetic code that makes up their genome. These genomes differ from species to species in ways such as genome size, GC content, or ability to multiply P-rich rRNAs, among others. The full expression of these genome 'traits' requires energy and block-building materials, and therefore they can determine the presence of a species in a given site. The authors describe the results of an experiment in which nutrient-poor springs (oligotrophic conditions) were experimentally NP fertilized (creating a copiotrophic environment). The response of microbial communities in these lakes, assayed using state-of-the-art metagenomic methods, conformed to predictions. This work is unique in that it merges ideas gathered from trait-based macroorganism ecology to microbial communities in whole-lake ecosystems. By doing so, the authors make use of, rather for the first time, fundamental genome-wide traits as a determinant of species composition.

**Decision letter after peer review:**

Thank you for submitting your article "Genomic adaptations in information processing underpin trophic strategy in a whole-ecosystem nutrient experiment" for consideration by *eLife*. Your article has been reviewed by two peer reviewers, and the evaluation has been overseen by a Reviewing Editor and Detlef Weigel as the Senior Editor. The reviewers have opted to remain anonymous.

The reviewers have discussed the reviews with one another and the Reviewing Editor has drafted this decision to help you prepare a revised submission.

Summary:

Okie et al. describe the results of an experiment in which nutrient poor springs were experimentally supplemented with a limiting nutrient. The response of microbial communities was assayed using metagenomic methods. In particular, microbial genomic traits were inferred from the metagenomic data, and related to organismal life history. The study has an overarching goal of linking 'trait-based' approaches that have been highly useful in non-microbial ecology to a meta-genomic investigation, which is an important approach to the study of microbial communities. The sampling and field experiment appear to reviewers to be technically sound, allowing for practical constraints that are described and justified. The data processing workflows follow standard practices up to a point – some analyses address novel questions, for which there are no established 'best practice' workflows, but the analytical choices seem reasonable. In sum, the manuscript by Okie et al. could be an interesting and worthwhile contribution to the literature, as it explores novel and interesting ideas, with a relatively strong dataset.

Essential revisions:

1) Is it possible the results are substantially affected by a 'bloom' in one or a few lineages in the P supplemented environments, and if so, does this matter? A hypothetical example might be a substantial increase in the abundance of a particular type of α-proteobacterium, with genome size and GC content (etc) greater than the community average. This kind of possibility (increases in prevalence of one or few particular 'species') would seem to accord quite well with the observed decrease in trait variance. Note, if the observed results were driven partially by growth responses of a few organisms, it would not affect the interpretation of the results too much. It might slightly increase the possibility that the outcomes are attributable to particular organisms, rather than a general response of organisms with certain traits responding in a way that holds generally. This would also affect the notion that decreased variance in a trait is an indication of how much that trait structures the community (e.g. subsection “Variance changes in the genomic traits”, first paragraph).

2) It's often difficult to replicate whole ecosystem experiments or to find appropriate controls. In this case, the authors used enclosures in the lake that were not fertilized. They compared samples from enclosures to the rest of the lake, which were not enriched. The authors pre-emptively try to address issues that might be raised by readers, and conclude that the only really difference is size (i.e., volume) between the non-fertilized and fertilized treatments. I wonder why the authors didn't use enclosures for both the control (non-fertilized) and enriched (fertilized) treatments. The authors argue that whole ecosystem experiments are important because mesocosms do not necessarily reflect observations that are relevant to the whole ecosystem. While this is a defensible argument it is somewhat diminished by the fact that their experiment inherently relies on mesocosms. In fact, one has to believe that the mesocosms are an accurate representation of the whole lake if the results are to be taken at face value. Also, reviewers were not sure the paper makes a compelling case that a whole-ecosystem approach was required to test hypotheses about how genomic attributes change in response to nutrient enrichment. In the end, reviewers were sympathetic to the constraints and applaud the ambition to tackle such questions at the ecosystem scale. Nevertheless, it is hard to completely overlook some of the limitations of the design. Please make sure these and other caveats are properly addressed in the Discussion section.

3) Reviewers were motivated by your questions and therefore were excited to read the manuscript. They were a bit less enthusiastic about how the paper came together in terms of organization and narrative. One of them said “the paper was at least a few drafts away from being "polished", It felt as though the manuscript would have benefited from some additional editing and revision”. So, please make sure the theory and arguments are be tightened up a bit, which will likely improve the flow of the paper.

---

## [Author Response]

Essential revisions:1) Is it possible the results are substantially affected by a 'bloom' in one or a few lineages in the P supplemented environments, and if so, does this matter? A hypothetical example might be a substantial increase in the abundance of a particular type of α-proteobacterium, with genome size and GC content (etc) greater than the community average. This kind of possibility (increases in prevalence of one or few particular 'species') would seem to accord quite well with the observed decrease in trait variance. Note, if the observed results were driven partially by growth responses of a few organisms, it would not affect the interpretation of the results too much. It might slightly increase the possibility that the outcomes are attributable to particular organisms, rather than a general response of organisms with certain traits responding in a way that holds generally. This would also affect the notion that decreased variance in a trait is an indication of how much that trait structures the community (e.g. subsection “Variance changes in the genomic traits”, first paragraph).

We found that a variety of taxa changed in abundance in response to fertilization (see Appendix 1—Figures 3 and 4 and newly added text in Appendix 1, subsection “Results – Changes in Taxonomic Composition”), so our results were clearly not driven by a bloom of just a few particular species. We added statistics and text towards the beginning of the Results to strengthen our answer to these points.

Note that some traits showed an increase in variance with fertilization, so we believe it is very unlikely that our variance results could be explained by just a few species responses.

It is also remarkable that some genomic traits, such as GC content, are not strongly phylogenetically conserved. For example, in α-proteobacterium, GC content varies from less than 30% to greater than 60% (Hildebrand et al., 2010). Thus, even if it were the case that just one phyla responded to nutrient enrichment (which it is important to stress was not the case), the question of which species within the phyla responded and the role of their traits in governing these responses would still be a relevant question.

2) It's often difficult to replicate whole ecosystem experiments or to find appropriate controls. In this case, the authors used enclosures in the lake that were not fertilized. They compared samples from enclosures to the rest of the lake, which were not enriched. The authors pre-emptively try to address issues that might be raised by readers, and conclude that the only really difference is size (i.e., volume) between the non-fertilized and fertilized treatments. I wonder why the authors didn't use enclosures for both the control (non-fertilized) and enriched (fertilized) treatments. The authors argue that whole ecosystem experiments are important because mesocosms do not necessarily reflect observations that are relevant to the whole ecosystem. While this is a defensible argument it is somewhat diminished by the fact that their experiment inherently relies on mesocosms. In fact, one has to believe that the mesocosms are an accurate representation of the whole lake if the results are to be taken at face value. Also, reviewers were not sure the paper makes a compelling case that a whole-ecosystem approach was required to test hypotheses about how genomic attributes change in response to nutrient enrichment. In the end, reviewers were sympathetic to the constraints and applaud the ambition to tackle such questions at the ecosystem scale. Nevertheless, it is hard to completely overlook some of the limitations of the design. Please make sure these and other caveats are properly addressed in the Discussion section.

Thank you for thoughtfully drawing attention to these complex issues. In this new version of the manuscript. we made additions towards the beginning of the Discussion to further address these questions and added further explanation in the Materials and methods that also address your concerns.

Our main idea and logic is that there is an asymmetry in the usefulness of internal fertilized and unfertilized mesocosms. Perturbation of a community (here the planktonic microbial community) usually causes local extinctions, opening up resources and niches for other species to colonize the local communities. From other studies in the same system we know that the pool of species that may potentially be an important source of colonizers in Lagunita pond water is the community of microbes living on the shores of the pond and in the sediments. Unlike a whole-ecosystem nutrient perturbation experiment, fertilizing a mesocosm would not sufficiently replicate many of these ecological and colonization processes that follow a nutrient perturbation – a mesocosm has no natural shore, and in the mesocosm there is decreased mixing between water column and sediments, reducing the maximum rate at which microbes from the sediment can colonize the water column. Since this study aims to understand the role of traits in the assembly of communities (not just the disassembly), it was important that potential colonizers not be artificially cut-off from the fertilized treatment and that is one of the main reasons for our design. We provided a summary of this logic in the Materials and methods, in addition to the other experimental design considerations explained in the Materials and methods.

3) Reviewers were motivated by your questions and therefore were excited to read the manuscript. They were a bit less enthusiastic about how the paper came together in terms of organization and narrative. One of them said “the paper was at least a few drafts away from being "polished", It felt as though the manuscript would have benefited from some additional editing and revision”. So, please make sure the theory and arguments are be tightened up a bit, which will likely improve the flow of the paper.

Thank you for the concerns and efforts to improve the readability and clarity of our study. We carefully revised the entire manuscript to improve the flow and tighten the concepts, arguments and description of the theory, including removing some unnecessary text in the Introduction and Conclusion, add clarifications to the predictions in the Introduction, strengthening the main message in the Discussion, and improving the organization of the Conclusion.

Thank you again, as we believe the polished paper is stronger and more cohesive, with a clearer logical flow and structure.